# Pretreatment expression of miR-191a may predict response to the induction chemotherapy based on cytarabine in acute myeloid leukemia patients – a single-center pilotal study

**Agnieszka Szymczyk**[1]*, **Sylwia Chocholska**[2], **Katarzyna Radko**[2], **Marek Hus**[2], **Monika Podhorecka**[2]

**1** Department of Hematology, National Medical Institute of the Ministry of Interior and Administration, Warsaw, Poland, **2** Department of Haematooncology and Bone Marrow Transplantation, Medical University of Lublin, Lublin, Poland

* agnieszka.szymczyk.med@wp.pl

## Abstract

### Background

Acute myeloid leukemia (AML) is associated with the accumulation of acquired genetic disorders. Moreorver chromosomal and molecular changes are independent prognostic factors that are taken into account to determine prognosis and treatment. MicroRNAs are novel gene regulators, which have been recognized to play an important role in pathological leukemogenesis. This study is aimed to analyze the possible role of micro RNAs as a markers predicting the outcome to induction chemotherapy based on cytarabine.

### Materials and methods

The expression of miR-34a, miR-191, miR-199a and miR-199b in previously separated bone marrow cells was assessed at the moment of diagnosis in 44 AML patients with use of qRQ-PCR technique. Assessment of response to induction therapy was based on criteria for response to treatment proposed by European LeukemiaNet (ELN).

### Results

Only the expression level of miR-191a out of all analyzed microRNAs was significantly associated with the induction chemotherapy response. We detected also significantly higher expression of miR-191 in *FLT3-ITD* negative group in comparison to *FLT3-ITD* positive subjects. For miR-34a, miR-199a and miR-199b, no relationship was found between their expression and *FLT3-ITD*, *NPM1* and *CEBPA* mutations. It

**Data availability statement:** All relevant data are within the article and its supporting information files.

**Funding:** This work was supported by the Medical University of Lublin under Grant Number 176.

**Competing interests:** The authors have declared that no competing interests exist.

was also shown that in the group of patients with low miR-191 expression, the number of myeloblasts was higher (p < 0.05).

## Conclusions

These results may prove an important role of miR-191a expression as a predictor of response to the induction chemotherapy based on cytarabine, even in cases when other risk factors are absent.

## Introduction

Acute myeloid leukemia (AML) accounts for about 80% of all acute leukemias diagnosed in adults. In Europe, its incidence is estimated at 5–8 cases per 100,000 individuals per year with median age at diagnosis of 67 years of age. [1,2]. AML is characterized by excessive proliferation and accumulation of immature blast cells derived from a transformed precursor myeloid cell. It is associated with the accumulation of acquired genetic disorders resulting primarily from exposure to environmental factors, such as: chemical substances (aromatic hydrocarbons), ionizing radiation, smoking, drugs [3].

The current classification of acute leukemia according to the World Health Organization (WHO) takes into account both chromosomal and molecular changes [4]. Cytogenetic changes are recognized as independent prognostic factors essential for determining prognosis and treatment. Approximately 55% of AML patients exhibit chromosomal aberrations detectable via the GTG (G-bands by trypsin using Giemsa) banding technique. The FISH (fluorescence in situ hybridization) technique further enables the detection of complex or cryptic translocations, marker chromosomes, and numerical abnormalities. [5]. A complex karyotype that is associated with an unfavorable clinical course and the associated low percentage of remissions to induction treatment and short overall survival (OS) is found in approximately 10% of patients with AML [6].

In addition to changes visible under a microscope, numerous submicroscopic changeshave also been identified. Among these, the internal tandem duplication of the FLT3 gene (FLT3-ITD) is a particularly significant prognostic factor, which is recommended for assessment in everyday clinical practice. Other markers with adverse prognostic value include WT1 mutations and overexpression of the *BAALC*, *ERG*, and *MN1* genes. However, presence of *CEBPA* and *NPM1* gene mutations are considered favorable prognostic factors. Based on molecular and cytogenetic changes, European LeukemiaNET (ELN) proposed a molecular cytogenetic classification in which three risk groups were identified: favorable, intermediate and unfavorable [7]. Additionally, patient-dependent factors, such as: age, presence of concomitant diseases assessed according to HCT-CI (Hematopoietic Cell Transplantation - Comorbiditi Index), general condition assessed according to ECOG (Eastern Cooperative Oncology Group) - further influence prognosis [8].

Despite the progress in understanding the pathophysiology of AML, many molecular mechanisms responsible for the disease are still not sufficiently understood.A growing body of evidence supports the complexity of the cytogenetic and molecular landscape in AML. Amon the emerging areas of research is the role of microRNAs (miRNAs) in leukemogenesis. MicroRNAs are epigenetic regulators that modulate gene expression and cellular signaling pathways.They may be deregulated in human cancers, some of them induce tumor growth, while others act as tumor suppressors. They expression can be used to predict prognosis and clinical response to treatment in cancer patients [9,10].

The aim of the presented study was to analyze a relationship between the expression of the chosen microRNA: miR-34a, miR-191, miR-199a and miR-199b and the response to induction chemotherapy in AML patients in the context of recognized prognostic factors (including *FLT3-ITD*, *FLT3D835*, *NPM1* and *CEBPA* mutations and cytognenetic changes) as well as parameters of complete blood count. The microRNA expression detected in bone marrow cells was evaluated in group of 44 adult patients in the moment of diagnosis just before starting an induction therapy. Assessment was performed with use of qRQ-PCR technique. The study was a single-center one, and the obtained results may be the basis for wild-range clinical trial.

## Materials and methods

### Study group characteristic

The study group included 44 patients diagnosed with AML treated in Department of Haematooncology and Bone Marrow Transplantation of Medical University of Lublin. The majority of patients were female (56.8%). The median age at diagnosis was 54.5 years (range: 18–82 years). In most patients (54.5%), the Hematopoietic Cell Transplantation Comorbidity Index (HCT-CI) score ranged from 1 to 2; in 15.9% of cases, the score was 0, while in 27.3% it was greater than 3. Patients' characteristic is presented in Table 1.

Clinical data and biological materials were collected from 19.12.2016 to 31.10.2018. Diagnosis was based on the morphological, immunophenotypic and cytogenetic criteria according to the World Health Organization (WHO) classification.

All patients enrolled in the study did not receive antileukemic treatment before sample collection. Exclusion criteria were: serious comorbidities (severe renal failure, autoimmune diseases, connective tissue systemic diseases, coexisting neoplastic diseases, severe heart failure, liver failure and/or respiratory failure), poor general condition according to Eastern Cooperative Oncology Group (ECOG≥2) as well as any previous treatment affecting the function of the immune system.

### Ethics approval and consent to participate

This study was approved by the Ethics Committee of the Medical University of Lublin (REC number: KE-0254/342/2016). All studies were performed in accordance with the ethical standards of the Declaration of Helsinki and written informed consent was obtained from all patients.

### MicroRNA expression analysis

Total RNA enriched with small RNAs (<200 nt) was isolated from bone marrow cells using a mirVana miRNA Isolation Kit (Ambion, USA) according to the manufacturer's instructions. RNA purity and concentration were assessed spectrophotometrically. RNA purity was evaluated based on the A260/A280 absorbance ratio, which ranged from 1.8 to 2.0. The prepared RNA samples were stored at −80°C until further analysis. TaqMan MicroRNA RT Kit (Applied Biosystems, USA) was used for reverse transcription (RT). The reaction was carried out on a matrix of previously isolated RNA with miRNA-specific stem-loop primers. The resulting complementary DNA (cDNA) was subjected to further analytical procedures.

Relative quantitative analysis of microRNA expression was carried out using TaqMan MicroRNA Assay Kits and the Applied Biosystems 7300 Real Time PCR Instrument (Applied Biosystems, USA). Data were normalized to endogenous control expression, analysed using the threshold cycle (Ct) and presented as $2^{-\Delta Ct}$. Delta Ct (ΔCt) is the difference between the Ct of the target gene (CtT) and the endogenous control gene (CtE) (ΔCt = CtT − CtE). Each measurement was performed in duplicate, and the mean of the two readings was used for statistical analysis.

**Table 1. Baseline clinical and laboratory characteristics of the study group.**

| Number of patients | | 44 |
|---|---|---|
| **Age** | median [years] | 54.5 |
| | minimum - maximum | 18-82 |
| **Gender** | female | 25 |
| | male | 19 |
| **HCT-CI** | 0 | 7 |
| | 1-2 | 24 |
| | ≥3 | 12 |
| | unknown | 1 |
| **FLT3-ITD** | positive | 10 |
| | negative | 34 |
| **FLT3D835** | positive | 1 |
| | negative | 43 |
| **NPM1** | positive | 9 |
| | negative | 34 |
| | unknown | 1 |
| **CEBPA** | positive | 9 |
| | negative | 15 |
| | unknown | 20 |
| **Group of risk according to European Leukemia Net** | favorable | 17 |
| | intermediate | 4 |
| | unfavorable | 15 |
| | unknown | 8 |
| **WBC at diagnosis [G/l]** | medium ± SD | 56.4 ± 90.6 |
| | minimum - maximum | 0.1 - 441.3 |
| **Hgb at diagnosis [mg/gl]** | medium ± SD | 8.8 ± 1.7 |
| | minimum - maximum | 3.7 - 12.4 |
| **PLT at diagnosis [G/l]** | medium ± SD | 86.8 ± 168.1 |
| | minimum - maximum | 4 - 1109 |
| % of myeloblasts at diagnosis | medium ± SD | 64.3 ± 24.0 |
| | minimum - maximum | 20 - 98 |
| **Induction therapy based on cytarabine (n)** | DAC (daunorubicin, cytarabine, cladribine) | 23 |
| | DA (daunorubicin, cytarabine) | 3 |
| | 2CdA+AraC (cladribine, cytarabine) | 5 |

## Assessment of response to treatment and survival in the study group

The study group responded to induction treatment based on European LeukemiaNet 2022 (ELN22) recommendations and estimated survival function assessing probability of overall survival (pOS), probability of progression-free survival (probability of progression) free survival, pPFS). CR was defined as the presence of ≤ 5% blasts in bone marrow, absence of circulating blasts and extramedullary disease, absolute neutrophil count (ANC) ≥ 1.0 G/l, platelet count (PLT) ≥ 100 G/l [7]. PFS was defined as the period of time from diagnosis to relapse, treatment failure, or death (from any cause). OS was defined as the period of time from diagnosis to death.

**Induction therapy**

Of the 44 patients included in the study, 31 received intensive induction chemotherapy—26 patients were treated with the DAC regimen (daunorubicin, cytarabine, and cladribine), and 3 with the DA regimen (daunorubicin and cytarabine). Nine patients received non-intensive treatment regimens. One patient died before treatment initiation, and treatment data were unavailable for 3 patients.

Assessment of response to induction therapy was based on criteria for response to treatment proposed by European LeukemiaNet 2022 (ELN22). Response to therapy was observed in 24 subjects - including 21 patients treated with intensive regimens and 10 with non-intensive regimens. No response was noted in 13 patients (29.1%) - including 7 receiving intensive therapy and 6 on non-intensive protocols. Remission was confirmed by flow cytometry and defined as the presence of $<10^{-3}$ residual blasts in the bone marrow.

**Statistical analysis**

Statistical analysis of the obtained results was carried out using Statistica 12.0 for Windows (StatSoft, USA). The D'Agostino–Pearson test was used to assess the normality of distribution for continuous variables. Since the data did not meet the assumptions of a normal distribution, the median was used as a measure of central tendency, and the interquartile range and/or minimum–maximum range were used to describe variability.

The expression levels of the analyzed genes were categorized as low or high based on whether the values were below, equal to, or above the median.

To compare continuous variables - including the expression of the studied miRNAs based on demographic and clinical factors, as well as clinical-demographic, morphological, genetic, and epigenetic factors depending on remission status the Mann-Whitney U test (for two groups) or the Kruskal–Wallis test (for more than two groups) was applied. For categorical variables, comparisons between groups were performed using Fisher's exact test (for two-category variables across two groups) or the chi-square test with Yates' correction (when the number of variable categories or groups exceeded two). Correlations between selected clinical-demographic variables, morphological parameters, and miRNA expression levels were analyzed using Spearman's rank correlation test. The diagnostic utility of the examined miRNAs in predicting remission was evaluated using receiver operating characteristic (ROC) curves.

Univariate analysis of the risk of shortened time to disease relapse was conducted using the log-rank test, with results reported as hazard ratios (HR) along with 95% confidence intervals (CI).Multivariate analysis of the association between selected clinical-demographic, genetic, and epigenetic variables and the risk of earlier relapse was conducted using Cox proportional hazards regression models. All variables found to be statistically significant in the univariate analysis were included in the multivariate models. A backward elimination method was applied, which confirmed their relevance, and the final models were adjusted accordingly. In all analyses, a p-value <0.05 was considered statistically significant. The results were presented numerically, graphically and descriptively.

## Results

### Assessment of microRNA expression

The results of MiR-34a, miR-191, miR-199a, miR-199b expression assessed in group of 44 AML patients are shown in Table 2 and Suplemental material. We detected overexpression of miR-191 in 35 out of 44 subjects (79.5%). In remaining cases, expression was lower or equal to that of endogenous control. In 40 patients the level of miR-199a expression was comparable to expression of the endogenous control, while in 4 cases it was significantly lower - the $2^{-\Delta Ct}$ values were 0.0001 (n = 2) and 0.0002 (n = 2), respectively. The level of miR-199b was comparable to expression of the endogenous control in 11 patients (25.0%), and only in 1 subject (2.3%) in case of miR-34a.

**Table 2. Analysis of $2^{-\Delta\Delta Ct}$ values of the studied microRNAs in the studied group of patients with AML.**

| microRNA | M±SD | minimum | maximum |
|---|---|---|---|
| miR-34a | 0.0841±0.1528 | 0.0037 | 1.000 |
| miR-191 | 1.7865±1.0503 | 0.0627 | 4.42320 |
| miR-199a | 0.9091±0.2907 | 0.0001 | 1.0000 |
| miR-199b | 0.3416±0.4789 | 0.0001 | 1.0000 |

M, mean; SD, standard deviation.

### Analysis of microRNA expression in relation to complete blood cell count parameters

A statistically significant positive correlations between lymphocyte and basophil counts and miR-199a expression were detected ($p < 0.05$). Negative correlation between MPV values and miR-199a and miR-199b expression were found ($p < 0.05$). As far as the number of myeloblasts was concerned the statistically significant negative correlation with miR-191 expression was observed ($p < 0.05$). No relationship was found when other parameters of CBC in relation to the expression of the studied microRNAs were analyzed.

### Assessment of microRNA expression in regard to the patient-dependent prognostic factors

The patient-dependent prognostic factors were assessed, such as age, general condition according to ECOG classification and the presence of co-morbidities according to HCT-CI. A statistically significant positive correlation was found between age and miR-34a expression. In addition, the group of younger patients (<60 years of age) had a lower average expression of miR-34a (0.0585±0.0683), while in the group of older patients (> 60 years of age) it was significantly higher (0.1523±0.2694). There was no correlation between the studied microRNAs expression and the ECOG status in any of the analyzed cases. Similarly, the number of co-morbidities assessed according to HCT-CI and the associated risk of death not resulting from AML diagnosis did not affect the level of microRNAs expression.

### Analysis of relationship between microRNA expression and genetic abnormalities

There were no differences in the expression of microRNAs in the cytogenetic-molecular risk groups assessed according to European Leukemia Net, when expression of particular microRNA was compared in group of favorable, intermediate and unfavorable risk, respectively. Thus, further analysis focused on comparing microRNAs expression in regard to molecular changes such as FLT3-ITD, FLT3D835, NPM-1 and CEBPA mutations was performed. Due to the fact that only one patient had positive FLT3D835 expression, the relationship between its expression and microRNA expression was abandoned. In the case of miR-34a, miR-199a and miR-199b, there was no statistically significant relationship between their expression and FLT3-ITD, NPM1 and CEBPA expression. However, a statistically significant correlation was detected in case of miR-191 expression and FLT3-ITD presence. In the group of patients FLT3-ITD negative the expression of miR-191 was significantly higher in comparison to FLT3-ITD positive subjects. The data is shown in Table 3.

Analysis of the diagnostic utility of the studied microRNAs in detecting disease remission. Univariate analysis did not show any statistically significant diagnostic utility of the studied microRNAs in detecting remission. Detailed data from this analysis are presented in Table 4.

Association between microRNA expression and progression-free survival (PFS). Univariate analysis did not show a significant association between the expression levels of the analyzed microRNAs and PFS in the study group (Table 5).

Similarly, the multivariate analysis did not reveal a statistically significant association between the expression levels of the analyzed microRNAs and progression-free survival (PFS).

Detailed data are presented in Table 6.

**Table 3. Assessment of miR-34a, miR-191, miR-199a and miR-199b expression in the context of response to induction chemotherapy and selected prognostic factors.**

**a. Evaluation of microRNAs expression in regard to FLT3-ITD expression.**

| microRNA | FLT3-ITD | | |
|---|---|---|---|
| | M±SD | | p |
| | negative | positive | |
| miR-34a | 0.0937±0.1725 | 0.0514±0.0304 | 1.000 |
| miR-191 | 1.9754±1.0702 | 1.2050±0.7642 | 0.0321 |
| miR-199a | 0.9118±0.2778 | 0.9000±0.3161 | 0.9665 |
| miR-199b | 0.3537±0.4845 | 0.3005±0.4826 | 0.5015 |

b. Assessment of the impact of miR-34a, miR-191, miR-199b, FLT3-ITD and NPM-1 expression on induction chemotherapy response - logistic regression coefficient analysis.

| | B | SE | OR | 95% CI | | p |
|---|---|---|---|---|---|---|
| | | | | LL | UL | |
| Constant | -1,79 | 1,78 | | | | |
| miR-34a | 9,18 | 13,47 | 9663,370 | 0,000 | 2,81 * $10^{15}$ | 0,507 |
| miR-191 | 1,99 | 1,28 | 7,286 | 0,593 | 89,532 | 0,016 |
| miR-199b | -1,19 | 1,94 | 0,304 | 0,007 | 13,519 | 0,785 |
| FLT-ltd | -1,17 | 0,60 | 0,097 | 0,009 | 1,021 | 0,022 |
| NPM-1 | 1,38 | 1,10 | 15,854 | 0,215 | 1168,069 | 0,164 |

B - regression coefficient; SE- standard error; OR - odds ratio; CI - confidence interval; LL - lower limit; UL - upper limit; p - significance level

**Table 4. Analysis of the diagnostic utility of the studied microRNAs in detecting disease remission.**

| Zmienna | Remisja choroby | | | | |
|---|---|---|---|---|---|
| | Czułość (%) | Specyficzność (%) | Punkt odcięcia | AUC [95%CI] | p |
| Ekspresja miRNA-34a | 41,6 | 38,4 | >0,04 | 0,5 [0,3-0,6] | 1,0000 |
| Ekspresja miRNA-191 | 33,3 | 100 | >2,39 | 0,6 [0,4-0,8] | 0,0970 |
| Ekspresja miRNA-199b | 70,8 | 53,8 | >0,01 | 0,54 [0,3-0,7] | 0,6574 |

Association between microRNA expression and overall survival (OS). Univariate analysis did not reveal any association between the expression levels of the studied microRNAs and overall survival (OS) (Table 7). Likewise, multivariate analysis did not show any statistically significant associations (Table 8).

## Analysis of the impact of microRNAs expression on the response to AML induction therapy

The relationship between the expression of microRNAs and the overall response rate (ORR) was assessed based on logistic regression analysis using the highest likelihood method. Due to the number of all studied patients, only five parameters could be analyzed together. Thus, miR-34a, miR-191, miR-199b (level of expression in regard to control gene), as well as FLT3 and NPM-1 (presence or absence) were chosen as the variables introduced into the model. The constructed model did allow to predict the response to induction chemotherapy significantly better than the model taking into account only the constant [$\chi2$ (5) = 13.54; p=0.019]. Based on the analysis of R2 Nagelkerke (0.529), a fairly strong relationship

**Table 5. Association between microRNA expression levels and progression-free survival (PFS) in the study group (univariate analysis).**

| Variable | Analiza jednoczynnikowa | | |
|---|---|---|---|
| | Median PFS | HR | *p* |
| | | (mies.) | [95%CI] |
| **miRNA34a** | | | |
| High | 2.0 | 1.4 | 0.3211 |
| Low | 3.0 | [0,6-3,2] | |
| **miRNA191** | | | |
| High | 4.0 | 0.6 | 0.2809 |
| Low | 2.0 | [0.2-1.5] | |
| **miRNA199b** | | | |
| Low | 2.0 | 1.1 | 0.8041 |
| High | 3.0 | [0.4-2.8] | |

**Table 6. Association between the expression levels of analyzed microRNAs and progression-free survival (PFS) in the study group (multivariate analysis).**

| Variable | Multivariate analysis | |
|---|---|---|
| | HR | *p* |
| | | [95%CI] |
| **miRNA34a** | | 0,3609 |
| High | 1,6 | |
| Low | [0,5-4,4] | |
| **miRNA191** | | 0,8675 |
| High | 1,1 | |
| Low | [0,2-4,3] | |
| **miRNA199b** | | 0,9421 |
| Low | 0,9 | |
| High | [0,3-3,0] | |

**Table 7. Analysis of the association between the expression levels of the studied microRNAs and overall survival (OS) (univariate analysis).**

| Variable | Univariate analysis. | | |
|---|---|---|---|
| | Median OS (Overall Survival). (month.) | HR | *p* |
| | | | [95%CI] |
| **miRNA34a** | | | 0.1675 |
| High | 2.0 | 1.6 | |
| Low | 3.0 | [0.7-3.8] | |
| **miRNA191** | | | 0.1621 |
| High | 6.0 | 0.5 | |
| Low | 2.0 | [0.2-1.4] | |
| **miRNA199b** | | | 0.7127 |
| Low | 3.0 | 1.1 | |
| High | 3.0 | [0.4-3.1] | |

**Table 8. Association between the expression levels of the analyzed microRNAs and overall survival (OS) in the study group (multivariate analysis).**

| Variable | Multivariate analysis. | |
|---|---|---|
| | HR | p |
| | | [95%CI] |
| miRNA34a | | 0.3015 |
| High | 1.6 | |
| Low | [0.6-4.1] | |
| miRNA191 | | 0.4707 |
| High | 0.7 | |
| Low | [0.3-1.8] | |
| miRNA199b | | 0.4417 |
| High | 1.4 | |
| Low | [0.5-4.0] | |

between prediction and patient distribution in responders and non-responders group. The prediction success was 95.5% for patients who responded to induction chemotherapy and 50% for patients who did not respond to treatment (Table 9). Therefore, it can be concluded that based on the expression of FLT3-ITD and NPM-1 as well as miR-34a, miR-191, miR-199b, the response to induction chemotherapy in AML patients can be predicted quite well.

Among the predictors introduced into the model, miR-191 and FLT-ITD were significantly stronger predictors of response to induction chemotherapy. An increase in miR-191 expression by 1 (expression level corresponding to the reference gene) increases the chance of responding to induction chemotherapy 7.228-fold (95% CI: 0.980–89.532). In turn, FLT-ITD expression reduces the chance of response 10.309-fold (95% CI: 0.593–108.16). Logistic regression coefficients are presented in Table 3.

### Sensitivity and specificity of expression of selected microRNAs in the field of predicting response to induction chemotherapy

In the next part of analysis, the sensitivity and specificity of the studied microRNA expression in terms of predicting the response to induction chemotherapy were calculated. For this purpose, a ROC curve was performed for miR-34a, miR-191, miR-199b. The expression of each of the studied microRNAs was treated as a stimulant of the distinguished state, which was the occurrence of a response to induction chemotherapy. Then the cut-off point was determined for each of the curves using the Youden index method.

Only the expression level of miR-191a out of all microRNAs was significantly associated with the induction chemotherapy response, as found by analyzing the area under the ROC curve (AUC). A miR-191 expression level of 1.9 above the reference gene expression was determined to be the cut-off point. A sensitivity of 54.5% and a specificity of 88.9% (Youden index of 0.434) were obtained for this value. ROC curves as well as the specificity of ROC curves and cut-off points for all analyzed miRNAs are shown in Fig 1.

**Table 9. Effectiveness of the model in predicting response to DAC induction therapy based on the parameters included in the model.**

| | | Predicted results | | % |
|---|---|---|---|---|
| | | yes | no | |
| Achieved results | yes | 21 | 1 | 95.5 |
| | no | 4 | 4 | 50.0 |

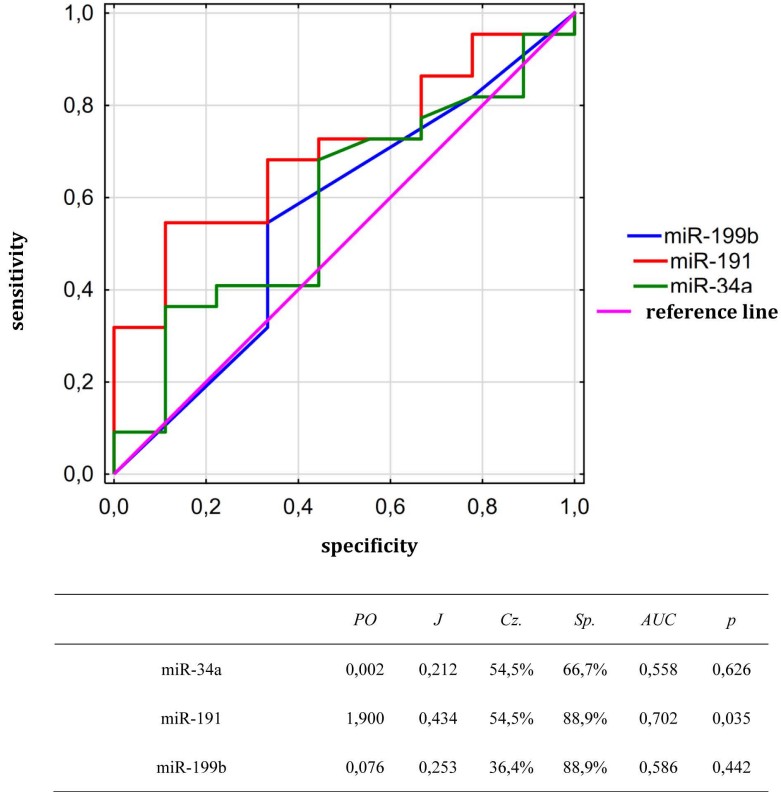

| | PO | J | Cz. | Sp. | AUC | p |
|---|---|---|---|---|---|---|
| miR-34a | 0,002 | 0,212 | 54,5% | 66,7% | 0,558 | 0,626 |
| miR-191 | 1,900 | 0,434 | 54,5% | 88,9% | 0,702 | 0,035 |
| miR-199b | 0,076 | 0,253 | 36,4% | 88,9% | 0,586 | 0,442 |

PO - cut-off point; J - Youden index; Vol. - sensitivity; Sp. - specificity; AUC - area under the curve;

p - significance

**Fig 1. Comparison and specificity of ROC curves for evaluated microRNAs and proposed cut-off points for the studied microRNAs in the context of predicting response to induction chemotherapy.**

## Discussion

One of the main restrictions on the use of chemotherapy in patients with AML is drug resistance, which is explained by a number of cellular and biochemical mechanisms. Among them reduction of cellular activation or deactivation, intensification of repair processes of DNA damage caused by cytostatic drugs or lack of recognition, impaired drug accumulation inside the cell, impaired its activation within the cell or increased intracellular catabolism, disturbances in the process of apoptosis, changes in the function, concentration or activity of the target may be of importance. Drug resistance of leukemia cells is influenced by cytogenetic and molecular aberrations, both primary and selected during the development of leukemia. Numerous scientific reports indicate a relationship between the occurrence of cytogenetic and molecular aberrations and resistance to the induction chemotherapy used [11]. Literature data also confirm relationship between the time required to enter complete remission and overall surviva (OS) and progression free survival (PFS) [12].MicroRNAs are novel gene regulators, which have been recognized to play an important role in pathological leukemogenesis due to their oncogenic functions and tumor suppressive [13]. The role played by microRNAs in the regulation of gene expression of enzymes responsible for drug metabolism (e.g., cytochrome P450 3A4), drug transporters like (e.g., breast cancer resistance protein, BCRP) or other drug targets is also significant [14].

In the presented study we tried to analyze the possible role of chosen micro RNAs expression as markers predicting the outcome of induction chemotherapy based on cytarabine in AML patients. First, the relationship between microRNAs

expression and recognized prognostic factors and as well as parameters of complete blood count were analyzed. We detected a positive correlation between miR-34a expression and the age of the subjects, but no relationship was found between the general condition of patients assessed according to ECOG and the number of comorbidities assessed according to HCT-CI and the level of examined microRNAs. A positive correlation was found between lymphocyte and basophil counts and miR-199a expression, and a negative correlation between MPV values and miR-199a and miR-199b expression. It was also shown that in the group of patients with low miR-191 expression, the number of myeloblasts was higher. Among the patients of the analyzed group there were no differences in the expression of microRNAs in individual cytogenetic-molecular risk groups assessed according to European LeukemiaNET. However, further analysis regarding molecular abnormalities revealed significant correlations. Higher expression of miR-191 was detected in *FLT3-ITD* negative group in comparison to *FLT3-ITD* positive subjects. For miR-34a, miR-199a and miR-199b, no relationship was found between their expression and *FLT3-ITD*, *NPM1* and *CEBPA* mutations.

The relationship between miR-34a expression and cardiac aging has been reported [15], but there are no such data for patients with haematopoietic proliferative diseases. There are also no reports regarding the relationship between other non-genetic prognostic factors and parameters of complete blood count and the expression of miR-34a, miR-191, miR-199a and miR-199b in patients with AML, although literature data report this type of correlation regarding other microRNAs. Zhao J. et al. [16] describes the negative correlation between white blood cell (WBC) and myeloblasts and negative correlation between hemoglobin (Hgb) and platelet (PLT) count and miR-96 expression. Xu LH. et al. [17] instead, showed positive correlation between miR-155 expression and WBC, serum lactate dehydrogenase (LDH) activity, C-reactive protein (CRP) level and miR-196b and miR-25 expression. Compared to miR-34 family, which is part of the type of conserved microRNAs, the relationship between its expression and AML development has already been proven. Literature data report that a decrease in miR-34a expression results in dysregulation of the p53 suppressor protein, and thus leads to disorders of the apoptosis process. AML patients with miR-34a overexpression most often have better prognosis [9]. Similarly, results published by Liu X. et al. [18] confirmed that patients with AML (both newly diagnosed, recurrent cases and after induction treatment) had higher miR-34a expression than those in the general population. Brishe et al. [19] had demonstrated that miR-34a expression can predict the response to cytarabine based therapy in AML patients. Additionally, the authors have shown that miR-34a-5p and miR-24-3p regulate deoxycytidine kinase (DCT), which is an enzyme involved in activation of cytarabine [19].

Literature data on miR-199 and miR-191 expression in AML patients is scarce. Favreau A. et al. [20] proved that loss of miR-199b can lead to pathological myeloproliferation, in addition low expression miR-199b in AML patients correlates with worse OS. Garzon et al. [21] demonstrated that a subset of miRNAs is clearly deregulated in AML and associated with cytogenetic groups and outcome in comparison to healthy persons. Although this study was not designed to answer whether miRNAs could predict outcome, the authors additionally analyzed a miRNA signature significantly associated with patients survival time. The results showed that subjects with high expression of *miR-191* and *miR-199a* had significantly worse overall and event-free survival than AML patients with low expression. Surprisingly, other characteristics, such as age, white blood cells, and *FLT3*-ITD were not significantly associated with OS nor EFS in this group of patients. Thus, that was clearly indicated by authors, further studies would be required to address the accuracy and independent prognostic significance of these miRNAs in predicting outcome. In contrast to the study presented by Garzon et al. [21] we focus on the role of micro-RNA expression in predicting the response to anti-leukemia induction therapy, while an impact on survival time was not measured. Our results indicates the significant role of miR-191 together with FLT-ITD as predictors of response to induction chemotherapy based on cytarabine. Additionally, an adverse correlation between miR-191 and blasts cells number, that was detected in our study, may further prove the positive role of high miR-191 as prognostic marker.

However, several limitations should be acknowledged. Our study cohort was small and clinically heterogeneous, and cytogenetic and molecular testing was performed in accordance with the European LeukemiaNet 2022 (ELN22)

classification. Applying next-generation sequencing (NGS) to assess genetic alterations, and comparing those results with miRNA expression profiles, could yield more comprehensive insights into their co-occurrence and potential influence on PFS and OS.

In the current study, the prognostic significance of miR-191 was confirmed only in a predictive model, and not through univariate or multivariate survival analysis. Our findings should therefore be considered preliminary and exploratory.

As a next step, it would be necessary to assess microRNA expression in a larger, more homogeneous patient cohort, ideally treated with a single standardized intensive chemotherapy protocol. At the same time, it should be emphasized that the findings regarding the impact of miR-191 expression on treatment response require further investigation.

## Supporting information

**S1 File. Supplemental material: MiR expression.**
(XLS)

## Author contributions

**Conceptualization:** Agnieszka Szymczyk, Sylwia Chocholska, Katarzyna Radko, Monika Podhorecka.

**Data curation:** Agnieszka Szymczyk, Sylwia Chocholska, Katarzyna Radko, Marek Hus, Monika Podhorecka.

**Formal analysis:** Agnieszka Szymczyk, Sylwia Chocholska, Katarzyna Radko, Monika Podhorecka.

**Funding acquisition:** Monika Podhorecka.

**Investigation:** Agnieszka Szymczyk, Monika Podhorecka.

**Methodology:** Agnieszka Szymczyk, Sylwia Chocholska, Katarzyna Radko, Monika Podhorecka.

**Project administration:** Agnieszka Szymczyk, Monika Podhorecka.

**Resources:** Agnieszka Szymczyk, Monika Podhorecka.

**Software:** Agnieszka Szymczyk, Monika Podhorecka.

**Supervision:** Agnieszka Szymczyk, Monika Podhorecka.

**Visualization:** Agnieszka Szymczyk, Monika Podhorecka.

**Writing – original draft:** Agnieszka Szymczyk, Monika Podhorecka.

**Writing – review & editing:** Agnieszka Szymczyk, Sylwia Chocholska, Katarzyna Radko, Marek Hus, Monika Podhorecka.

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
