## [Decision Letter · Decision Letter 0]

12 Feb 2025

PONE-D-24-51496Pretreatment expression of miR-191a may predict response to the induction chemotherapy based on cytarabine in acute myeloid leukemia patients – a single-center studyPLOS ONE

Dear Dr. Szymczyk,

Thank you for submitting your manuscript to PLOS ONE. After careful consideration, we feel that it has merit but does not fully meet PLOS ONE’s publication criteria as it currently stands. Therefore, we invite you to submit a revised version of the manuscript that addresses the points raised during the review process by both Reviewers.

We look forward to receiving your revised manuscript.

Kind regards,

Francesco Bertolini, MD, PhD

Academic Editor

PLOS ONE

Journal Requirements:

“This study was funded by the research grant of Medical University of Lublin [DS176]”

Reviewers' comments:

Reviewer's Responses to Questions

**Comments to the Author**

1. Is the manuscript technically sound, and do the data support the conclusions?

Reviewer #1: Partly

Reviewer #2: Partly

2. Has the statistical analysis been performed appropriately and rigorously? 

Reviewer #1: Yes

Reviewer #2: I Don't Know

3. Have the authors made all data underlying the findings in their manuscript fully available?

Reviewer #1: Yes

Reviewer #2: No

4. Is the manuscript presented in an intelligible fashion and written in standard English?

Reviewer #1: Yes

Reviewer #2: No

5. Review Comments to the Author

Reviewer #1: In this paper, the Authors report on the impact of pre-treatment expression of miRNA -191 on response rate in AML patients. They found that miRNA-191 positively correlated with achievement of CR and the presence of miRNA-191 was higher among FLT3-ITD negative patients. Conversely, FLT3-ITD was related to a reduced probability of CR.

The role of miRNAs in AML is still not completely elucidated and new informations are needed.

However, at the current stage, the paper has major issues that hamper pubblication.

Major

The cohort is small and heterogenous, patient receiving intensive anthracycline + citarabine induction are included alongside patients who received less intensive therapy without antracyclines. This may introduce a bias. Furhtermore, it is stated that 44 patients were enrolled but data on treatment received is reported in 31 (23 + 3 +5) patients only. Also, response rate are reported only for those 31 patients (22/31 achieved CR and 9/31 did not respond).

The Authors adopted the old ELN 2010 classification that included 4 risk group. I think that a newer edition (either 2017 or, better, 2022) should be adopted.

It is unclear how the variable for the predictive model were selected.

miRNA-191 is less frequent among FLT3-ITD negative patients and FLT3-ITD was related to a lower chance of CR (quite unusually, as this mutation usually does not impact CR rate in intensively treated patients but increased relapse risk). Can the Author try different predictive models without the inclusion of miRNA-191 or with other combination of variables in order to be certain that the effect is not mostly due to the presence or absence of FLT3-ITD.

Confidence intervals in the predictive model are quite wide and cross the unity (0.98-89.5 and 0.593 - 108.16 for miRNA-191 and FLT3-ITD, respectively). This is likely due to the small patient cohort.

The information would have been significantly more relevant if MRD data were provided

- the Author decided to omit survival analysis. I think it could have been useful.

Minor

there are several typos throughout the manuscript (e.g. NMP1 instead of NPM1).

Reviewer #2: Dear authors

The introduction is dense with information, which can make it difficult for readers to follow.

Some sentences are lengthy and contain multiple ideas, which may confuse readers.

Some information is repeated or could be consolidated.

The statistics provided regarding the incidence of AML could be presented more clearly. For example, stating the incidence rates in a comparative format would enhance clarity.

While the section states that patient characteristics are presented in Table 1, it does not provide any summary or highlights of these characteristics in the text.

The methodology for RNA isolation and analysis is described, but it lacks details about the quality control measures taken (e.g., RNA quality assessment) and how the specificity of the miRNA assays was ensured.

The statistical analysis section mentions using nonparametric tests due to the lack of normal distribution but does not provide any justification for the choice of tests used.

There is no justification provided for the sample size of 44 patients. A brief explanation of how this sample size was determined (e.g., power analysis) would strengthen the study’s methodology.

While the section states that patient characteristics are presented in Table 1, it does not provide any summary or highlights of these characteristics in the text.

There is little acknowledgment of potential methodological limitations within the study.

The discussion lacks a detailed description of the statistical methods used to analyze the correlations mentioned.

6. PLOS authors have the option to publish the peer review history of their article (what does this mean? ). If published, this will include your full peer review and any attached files.

**Do you want your identity to be public for this peer review?** For information about this choice, including consent withdrawal, please see our Privacy Policy .

Reviewer #1: **Yes: ** Fabio Guolo

Reviewer #2: No

---

## [Author Response · Author response to Decision Letter 1]

8 Apr 2025

Warsaw, 8th April 2025

Editor in Chief

PloseOne

Submission ID: PONE-D-24-51496

Title: Pretreatment expression of miR-191a may predict response to the induction chemotherapy based on cytarabine in acute myeloid leukemia patients – a single-center pilotal study

Dear Editor,

Please find our revised manuscript No. PONE-D-24-51496 entitled " Pretreatment expression of miR-191a may predict response to the induction chemotherapy based on cytarabine in acute myeloid leukemia patients – a single-center pilotal study". First, we would like to thank the Reviewers for their constructive evaluation of our manuscript. We found the comments valuable, and they helped to improve the manuscript. We prepared a revised version of the manuscript. New data are marked in the text.

The detailed (point-by-point) responses to the Reviewers’ comments are presented below.

Reviewer 1:

- The cohort is small and heterogenous, patient receiving intensive anthracycline + citarabine induction are included alongside patients who received less intensive therapy without antracyclines. This may introduce a bias. Furhtermore, it is stated that 44 patients were enrolled but data on treatment received is reported in 31 (23 + 3 +5) patients only. Also, response rate are reported only for those 31 patients (22/31 achieved CR and 9/31 did not respond).

Answer: Data on treatment response were not available in all cases. Some patients were lost to follow-up. The records were re-examined and missing data that were available were added to the article.

- The Authors adopted the old ELN 2010 classification that included 4 risk group. I think that a newer edition (either 2017 or, better, 2022) should be adopted.

Answer: We corrected it. We used the ELN2022 classification.

- It is unclear how the variable for the predictive model were selected.

Answer: We added this information.

- miRNA-191 is less frequent among FLT3-ITD negative patients and FLT3-ITD was related to a lower chance of CR (quite unusually, as this mutation usually does not impact CR rate in intensively treated patients but increased relapse risk). Can the Author try different predictive models without the inclusion of miRNA-191 or with other combination of variables in order to be certain that the effect is not mostly due to the presence or absence of FLT3-ITD.

Answer: Other models were tested.

- Confidence intervals in the predictive model are quite wide and cross the unity (0.98-89.5 and 0.593 - 108.16 for miRNA-191 and FLT3-ITD, respectively). This is likely due to the small patient cohort.

Answer: We have added information about the limitations of the study resulting from the small cohort.

- The information would have been significantly more relevant if MRD data were provided

Answer: Remmision was confirmed by flow cytometry and defined as < 10–3 residual blasts in bone marrow. We added this information.

- The Author decided to omit survival analysis. I think it could have been useful.

Answer: We added this information.

- There are several typos throughout the manuscript (e.g. NMP1 instead of NPM1).

Answer: We corrected typos.

Reviewer 2:

- The introduction is dense with information, which can make it difficult for readers to follow. Some sentences are lengthy and contain multiple ideas, which may confuse readers. Some information is repeated or could be consolidated.

Answer: We corrected it.

- The statistics provided regarding the incidence of AML could be presented more clearly. For example, stating the incidence rates in a comparative format would enhance clarity.

Answer: We added this information.

- While the section states that patient characteristics are presented in Table 1, it does not provide any summary or highlights of these characteristics in the text.

Answer: We added this information.

- The methodology for RNA isolation and analysis is described, but it lacks details about the quality control measures taken (e.g., RNA quality assessment) and how the specificity of the miRNA assays was ensured.

Answer: We added this information.

- The statistical analysis section mentions using nonparametric tests due to the lack of normal distribution but does not provide any justification for the choice of tests used.

Answer: We added this information.

- There is no justification provided for the sample size of 44 patients. A brief explanation of how this sample size was determined (e.g., power analysis) would strengthen the study’s methodology.

Answer: This is a pilot study. We used one kit for the microRNA analysis. We have added this information to the title of the article.

- While the section states that patient characteristics are presented in Table 1, it does not provide any summary or highlights of these characteristics in the text.

There is little acknowledgment of potential methodological limitations within the study.

Answer: We added this information.

- The discussion lacks a detailed description of the statistical methods used to analyze the correlations mentioned.

Answer: We added this information in Matherial and methods section.

We hope that the changes provided meet your expectations and that you will find our contribution suitable for publication in PloseOne.

Yours sincerely,

Agnieszka Szymczyk

---

## [Decision Letter · Decision Letter 1]

24 Apr 2025

Pretreatment expression of miR-191a may predict response to the induction chemotherapy based on cytarabine in acute myeloid leukemia patients – a single-center study

PONE-D-24-51496R1

Dear Dr. Szymczyk,

We’re pleased to inform you that your manuscript has been judged scientifically suitable for publication and will be formally accepted for publication once it meets all outstanding technical requirements.

Kind regards,

Francesco Bertolini, MD, PhD

Academic Editor

PLOS ONE

Additional Editor Comments (optional):

Reviewers' comments:

Reviewer's Responses to Questions

**Comments to the Author**

1. If the authors have adequately addressed your comments raised in a previous round of review and you feel that this manuscript is now acceptable for publication, you may indicate that here to bypass the “Comments to the Author” section, enter your conflict of interest statement in the “Confidential to Editor” section, and submit your "Accept" recommendation.

Reviewer #1: All comments have been addressed

2. Is the manuscript technically sound, and do the data support the conclusions?

Reviewer #1: Yes

3. Has the statistical analysis been performed appropriately and rigorously? 

Reviewer #1: Yes

4. Have the authors made all data underlying the findings in their manuscript fully available?

Reviewer #1: Yes

5. Is the manuscript presented in an intelligible fashion and written in standard English?

Reviewer #1: Yes

6. Review Comments to the Author

Reviewer #1: (No Response)

7. PLOS authors have the option to publish the peer review history of their article (what does this mean? ). If published, this will include your full peer review and any attached files.

**Do you want your identity to be public for this peer review?** For information about this choice, including consent withdrawal, please see our Privacy Policy .

Reviewer #1: **Yes: ** Fabio Guolo

---

## [Editor Report · Acceptance letter]

PONE-D-24-51496R1

PLOS ONE

Dear Dr. Szymczyk,

I'm pleased to inform you that your manuscript has been deemed suitable for publication in PLOS ONE. Congratulations! Your manuscript is now being handed over to our production team.

Kind regards,

on behalf of

Dr. Francesco Bertolini

Academic Editor

PLOS ONE